# Barley Leaf Ameliorates *Citrobacter rodentium*-Induced Colitis through Preventive Effects

**DOI:** 10.3390/nu14183833

**Published:** 2022-09-16

**Authors:** Yu Feng, Daotong Li, Chen Ma, Meiling Tian, Xiaosong Hu, Fang Chen

**Affiliations:** Key Laboratory of Fruits and Vegetables Processing, Ministry of Agriculture, Engineering Research Centre for Fruits and Vegetables Processing, National Engineering Research Center for Fruit and Vegetable Processing, College of Food Science and Nutritional Engineering, China Agricultural University, Beijing 100083, China

**Keywords:** barley leaf, colitis, *Citrobacter rodentium*, gut microbiota, prophylactic effect

## Abstract

The incidence and prevalence of inflammatory bowel disease (IBD) have been increasing globally and progressively in recent decades. Barley leaf (BL) is a nutritional supplement that is shown to have health-promoting effects on intestinal homeostasis. Our previous study demonstrated that BL could significantly attenuate *Citrobacter rodentium* (CR)-induced colitis, but whether it exerts a prophylactic or therapeutic effect remains elusive. In this study, we supplemented BL before or during CR infestation to investigate which way BL acts. The results showed that BL supplementation prior to infection significantly reduced the disease activity index (DAI) score, weight loss, colon shortening, colonic wall swelling, and transmissible murine colonic hyperplasia. It significantly reduced the amount of CR in the feces and also markedly inhibited the extraintestinal transmission of CR. Meanwhile, it significantly reduced the levels and expression of tumor necrosis factor-alpha (TNF-α), interferon-gamma (IFNγ), and interleukin-1β (IL1β). In addition, pretreatment with BL improved CR-induced gut microbiota dysbiosis by reducing the content of Proteobacteria, while increasing the content of *Lactobacillus*. In contrast, the effect of BL supplementation during infestation on the improvement of CR-induced colitis was not as good as that of pretreatment with BL. In conclusion, BL protects against CR-caused colitis in a preventive manner.

## 1. Introduction

Inflammatory bowel disease (IBD), a non-specific chronic recurrent inflammatory disease of the intestine, is characterized by abdominal pain and diarrhea, bloody stool, weight loss, and potential risk of cancer [1]. Epidemiological data indicated that the prevalence and incidence of IBD are increasing every year, and IBD has become a worldwide public health threat [2,3]. However, most of the current pharmacological treatments for IBD lack of specificity and may produce adverse effects such as exorbitant costs, immunosuppression, and drug resistance [4]. Therefore, developing a novel protective or therapeutic approach to IBD is of urgent need.

Although the precise pathogenesis of IBD is unclear, many studies have revealed that gut microbiota and eating habits perform crucial roles in the initiation and progression of IBD [5,6]. The mammalian gastrointestinal tract resides with trillions of microorganisms, and the makeup and diversity of the gut microbiota are pivotal for defending against pathogenic bacteria, regulating host immunity, and promoting nutrients’ absorption [7]. Previous studies indicate that the type, quality, and origin of food is the predominant factor influencing the structure of the gut microbiota, which, in turn, could affect host health and intestinal homeostasis [8,9]. Multiple studies have described variations in the intestinal flora structure between IBD patients and healthy individuals [10,11]. For example, the Enterobacteriaceae family, which contains many common Gram-negative pathogens, such as *Salmonella*, *Escherichia coli*, and *Shigella*, is elevated in IBD patients [10]. Meanwhile, there is a decrease in the amount of butyrate-producing intestinal bacteria in IBD patients [11]. Intriguingly, clinical trials have shown that donor fecal microbiota transplantation can effectively improve the symptoms of IBD patients and restore the intestinal flora structure of IBD to that of healthy individuals [12]. Thus, the regulation of gut microbiota might be a promising option for preventing and curing IBD.

*Citrobacter rodentium* (CR) is a natural mouse Gram-negative pathogenic bacterium that shares sixty-seven percent of its genes with human enteropathogenic *Escherichia coli* and enterohaemorrhagic *E. coli* [13]. Therefore, CR-induced bacterial colitis has long been utilized as a model to study the pathogenesis of enteric-infection-induced inflammation [14]. The infection of mice with CR triggers the formation of attaching and effacing (A/E) lesions, and this is distinguished by shedding of the brush border microvilli and intensive bacterial attachment [13]. Additionally, CR causes epithelial cell proliferation, colonic crypt hyperplasia, and mucosal thickening in mice. Following oral inoculation, CR causes a profound gut microbiota dysbiosis that is characterized by an overgrowth of Proteobacteria and a decline in the richness and overall diversity of the resident microbiota [14]. Initially, it colonizes the cecum patch, where it is considered to adapt to the intestinal environment in vivo. Then the pathogen migrates to the distal colon and undergoes rapid expansion by day 4 post-infection (p.i.). The CR burden peaks on day 7 p.i. and remains high up to day 12 p.i., and it is usually cleared by day 21 p.i. [14,15]. The capacity of the gut microbiota to colonize and outcompete CR is determined on nutritional availability [16], suggesting that dietary nutrients may serve as crucial factors to protect against CR-induced enteric infection.

Barley leaf (BL), the young leaf of *Hordeum vulgare* L., is a traditional Chinese herb that has been historically recorded to have potential beneficial health effects. The primary components of BL include dietary fiber, protein, polyphenol, vitamins, and minerals [17]. Multiple studies have revealed that BL exhibited antioxidant, lipid-lowering, and other physiological functions, which are effective in the treatment of several chronic disorders, such as diabetes and bone loss [18,19]. Yamaura et al. found that BL could partially inhibit the increase in NGF levels in the hippocampus and be antidepressant [20]. In addition, supplementation with BL should have health benefits by preventing diseases caused by oxidative damage, such as colorectal cancer and cardiovascular disease [21,22]. Our previous studies have demonstrated that dietary BL protects against CR-induced colitis (unpublished); however, whether it exerts a prophylactic or therapeutic effect remains elusive. In this study, we supplemented BL before (beCR) or during CR (duCR) infestation to evaluate the preventive or therapeutic impacts of BL on CR-induced colonic inflammation and gut microbiota dysbiosis.

## 2. Materials and Methods

### 2.1. Preparation of BL Powder

The powder of BL was prepared according to our previous studies [23,24]. Briefly, the fresh leaves of *Hordeum vulgare* L. (cultivated in Hangzhou, China) were cleaned, sliced, and dried. The dried BLs were then pounded into powder and filtered through a 300-mesh sieve. The nutritional composition of barley leaves used in this experiment is shown in Appendix A.

### 2.2. Animals

The mice utilized in this study were 4–6-week-old male C3H/HeN mice purchased from Beijing Vital River Laboratory Animal Technology Co., Ltd. (Beijing, China). After one-week of acclimatization, the mice were randomly divided into four groups: CD+CR, BL+CR, duCR, and beCR (*n* = 6 per group). CD+CR and BL+CR groups were fed a standard chow diet (CD) and isocaloric diet, in which 2.5% BL was added, respectively; the beCR group was fed BL chow before CR infection and switched to CD after infection, while the duCR group was fed CD before infection and BL chow at the time of infection. The macronutrient composition of the two diets is shown in Appendix A. Animal experiments were conducted following the National Institutes of Health guide for the care and use of Laboratory Animals (NIH Publications No. 8023, revised 1978), and the protocols were reviewed and approved by the Animal Care and Ethics Committee of China Agricultural University (Ethics reference number: AW32602202-4-2).

### 2.3. CR Infection

To induce bacterial colitis, mice were inoculated with approximately 1 × 10^9^ colony-forming units (CFUs)/mouse of CR strain DBS 100 (ATCC 51459). Briefly, a sterile Luria-Bertani medium was injected into an individual colony of CR grown on a fresh MacConkey agar plate (Solarbio, Beijing, China) at 37 °C and shaken overnight. After mice with CD and BL diet for 3 weeks, mice were infected.

### 2.4. Quantification of CR in Mouse Feces and Tissues

On days 1, 4, 7, and 10 post-infection, the fresh stool pellets of mice were harvested, weighed, and homogenized by using a BeadMill 24 benchtop bead-based homogenizer (Servicebio, Wuhan, China). Then sample homogenates were plated on MacConkey agar in a serial ten-fold dilution and CR colonies were counted during the following day. The CR colony has unique characteristics around a red center and a white edge, which could be clearly identified. Spleens and livers were collected and processed for organ bacterial-burden analysis.

### 2.5. Disease Activity Index (DAI)

The severity of colitis was evaluated by using the DAI score. Briefly, mouse DAI was calculated by weight loss (0 = none; 1 = 1–5%; 2 = 6–10%; 3 = 11–15%; 4 = >16%), stool consistency (0 = normal; 2 = loose stool; 4 = diarrhea), and general health status (0 = general well; 1 = slightly under average; 2 = poor; 3 = very poor; 4 = terrible), as previously described [25]. The maximum score was 12.

### 2.6. Histological Staining

The colonic and cecal tissues were embedded in paraffin after being fixed in 4% formalin solution. Following that, samples were sectioned and stained with hematoxylin and eosin (H&E). Tissue sections were estimated as previously described [25]. 

For goblet cell analysis, samples were stained with Alcian blue, using commercial kits (Servicebio Technology CO., Ltd., Wuhan, China). Image analysis was performed by utilizing Image J software.

### 2.7. Immunofluorescence Staining

The paraffin-embedded slices were deparaffinized and rehydrated before antigen retrieval with citric acid antigen retrieval solution (pH 6.0). Next, slides were blocked in goat serum. The primary antibody used was rabbit antisera generated against Ki67 (1:200; Abcam); after that, the CY3-conjugated goat anti-rabbit IgG (1:400; Jackson) as a secondary antibody was added to specially label the marker. Then sections were stained with 4′,6-diamidino-2-phenylindole (DAPI; Solarbio) and sealed with an anti-fluorescence quenching agent. The images were observed and collected under the Nikon fluorescence microscope.

### 2.8. Inflammatory Cytokine Analysis 

The enzyme-linked immunosorbent assay (ELISA) kit (Nanjing Jiancheng Institute of Biological Engineering, Nanjing, China) was applied to evaluate cytokine levels in the colon. First, ~100 mg colonic tissue was ground in liquid nitrogen and dissolved in tissue protein lysate (Solarbio, China). The supernatant was then centrifuged and measured for protein concentration, using a BCA protein assay kit (Solarbio, China). Afterward, IL-4, IL-1β, TNF-α, and IFN-γ were tested.

### 2.9. 16S rRNA Gene Sequencing

Extracted total DNA from mouse feces was utilized as a template for PCR amplification of the V3/V4 region of the bacterial 16S rRNA gene. PCR amplification products obtained by amplification on an ABI GeneAmp^®^ 9700 PCR system (AppliedBiosystems, Foster City, CA, USA) were sequenced on the Illumina Miseq platform in Majorbio BioPharm Technology Co., Ltd., (Shanghai, China) and further classified into operational taxonomic units (OTUs) within 0.03 (equivalent to 97% similarity) difference. The taxonomic community structure and phylogeny were assessed by visualizing datasets of microbial diversity and abundance of different samples. All data analyses were performed online on the Majorbio Cloud Platform. All raw sequence data were archived in the NCBI Short Read Archive database under the Bioproject accession number PRJNA870411.

### 2.10. Statistics

Data are presented as MEAN ± SEM. Statistical significance was determined by one-way or two-way analysis of variance (ANOVA), followed by the Duncan test, and the Bonferroni statistical test was applied post hoc. A *p*-value < 0.05 was considered statistically significant.

## 3. Results

### 3.1. Effect of beCR and duCR on CR-Induced Colitis

To investigate the preventive or therapeutic effects of BL on ameliorating CR-induced colitis, we designed the experiment in which mice were given BL before and during CR infection (Figure 1A). Compared with the CD+CR group, the mice in the BL+CR group exhibited significant improvement in body weight reduction, elevated DAI index, shortened colon length and swollen spleen due to CR infestation (Figure 1B–F). Notably, in comparison to mice supplemented with BL during CR infection, those supplemented with BL prior to infestation showed improvements in body weight loss and DAI severity, producing similar effects to the BL+CR group (Figure 1B,C). Despite the fact that there was no significant change in colon length between the duCR and beCR groups, the intestinal condition of mice in the beCR group was better than that in the duCR group, as shown by the presence of formed fecal pellets in the colon, improved intestinal wall swelling, and relieved cecum wrinkling (Figure 1D,E). Additionally, both beCR and duCR interventions could significantly improve splenomegaly due to CR, but no significant difference was found between them. Furthermore, there was no significant variation in diet and water consumption between groups, indicating that BL had no deleterious effects on the mice’s nutritional and water habits (Appendix A). Taken together, BL improvement of CR-caused enterocolitis might be largely in a preventive manner.

### 3.2. Effect of beCR and duCR on CR-Induced Intestinal Pathology

Transmissible murine colonic hyperplasia (TMCH) is the hallmark feature in CR-infected mice, exhibiting epithelial cell proliferation, colonic crypt hyperplasia, and mucosal thickening [14]. To investigate the effect of beCR and duCR on CR-induced intestinal pathology, we performed H&E staining, Alcian blue staining, and Ki67^+^ immunofluorescence staining on mouse colonic tissues. As compared with the CD+CR group, inflammatory cell infiltration, colonic crypt hyperplasia, and goblet cell deficiency were significantly alleviated in the colonic tissue of the BL+CR group mice (Figure 2A–E). The H&E results showed that beCR intervention could significantly improve the increase in mucosal thickness, reduce histopathological scores, and produce comparable intervention effects as BL+CR, whereas duCR intervention failed to alleviate the colonic crypt hyperplasia due to CR (Figure 2A–C). Ki67 is an associated antigen of immature proliferating cells and is essential in cell proliferation. Consistent with the H&E results, beCR intervention significantly ameliorated CR-induced goblet cell deficiency and Ki67^+^ cell overexpression, similar to the effect of BL+CR intervention; although duCR treatment reduced Ki67^+^ cell overproliferation, it was dramatically different from BL+CR and duCR intervention (Figure 2A,D,E). Furthermore, we performed H&E staining of the mouse cecum and observed that BL+CR and beCR interventions also significantly improved inflammatory cell infiltration, ulceration and loss of epithelial integrity in the cecum caused by CR, whereas duCR treatment failed to ameliorate these conditions (Appendix A). In conclusion, the beCR intervention was superior to the duCR intervention in improving mouse histopathology.

### 3.3. Effect of beCR and duCR on Inflammatory Cytokines in Colon

Cytokines are inflammatory regulators that are essential intermediary phenotypes in inflammatory diseases. Therefore, we detected the level of inflammatory cytokines in the colon by ELISA. As shown in Figure 3, dietary supplementation with BL did not have any impact on the content of the anti-inflammatory cytokine IL-4. However, BL+CR and beCR interventions remarkably decreased the concentrations of TNF-α, IL-1β, and IFN-γ, and the effects of both interventions were equivalent; meanwhile, the duCR intervention, although it reduced the elevated levels of pro-inflammatory cytokines due to CR, did not differ significantly from the CD+CR group. 

### 3.4. Effect of beCR and duCR on CR Colonization

To investigate the effect of beCR and duCR interventions on CR colonization in the intestine of mice, we collected mouse feces on days 1, 4, 7, and 10 after CR infection, and bacterial CFUs were enumerated by dilution and plate counts. On day 1 p.i., there was no significant difference in CR burden between groups of mice; however, on days 4, 7, and 10 p.i., beCR intervention significantly inhibited CR colonization in the intestine, producing an intervention effect similar to that of BL+CR; meanwhile, the CR burden in duCR-intervened mice was not significantly distinct from that in the CD+CR group (Figure 4A). Furthermore, we also examined the extraintestinal transmission of CR. As shown in Figure 4B,C, the beCR intervention significantly suppressed the amount of CR in the spleen and liver, exhibiting comparable effects to the BL+CR intervention, whereas the duCR intervention only significantly depressed the amount of CR in the spleen and had no significant effect on the amount of CR in the liver.

### 3.5. Effect of beCR and duCR on Modulating Alpha and Beta Diversity of Gut Microbiota

Numerous studies have demonstrated that the composition and structure of the gut microbiota perform an essential role in the progression and development of IBD [5,6]. Therefore, in this experiment, 16S rRNA sequencing was employed to investigate the effects of beCR and ducR on microbial community changes. Alpha diversity phenotypes the richness (Ace and Chao indexes) and diversity (Shannon and Simpson indexes) of the intestinal flora. In comparison to the CD+CR group, both BL+CR and beCR interventions significantly increased the ACE index and Chao index, indicating that both interventions could increase the richness of the intestinal flora; meanwhile, the duCR intervention had no effect on either index (Figure 5A,B). For improving gut microbiota diversity, the effect of beCR treatment was equivalent to that of BL+CR, whereas there was no significant difference between duCR and CD+CR groups (Figure 5C,D). The non-metric multidimensional scaling (NMDS) analysis results showed that mice in the BL+CR and beCR groups had a similar gut microbiota composition and structure; meanwhile, those in the duCR and CD+CR groups showed a comparable bacterial community in composition and structure, and this was consistent with the results of Partial-Least-Squares Discriminant Analysis (PLS-DA) analysis (Figure 5E,F).

### 3.6. Effect of beCR and duCR on Regulating Taxonomic Microbial Community Profiles

At the phylum level, dietary supplementation with BL could dramatically increase the abundance of Firmicutes, while suppressing the abundance of Proteobacteria (Figure 6A,B). At the genus level, in comparison with the CD+CR group, the beCR intervention could dramatically inhibit *Citrobacter* content, and the intervention effect was similar to that of BL+CR; meanwhile, in regard to the duCR intervention, although it could reduce *Citrobacter* content, it was not significantly distinguishable from the CD+CR group (Figure 6C–E and Appendix A). Notably, both the beCR and duCR intervention increased the abundance of *Lactobacillus* and did not differ significantly from the effect of BL+CR intervention, suggesting that BL might be a prebiotic enriched with *Lactobacillus* (Figure 6C–E and Appendix A). Moreover, the beCR treatment also increased the level of *unclassified_f_Lachnospiraceae*, *norank_f_ Lachnospiraceaeand*, and *Enterorhabdus*, and it decreased the level of *Erysipelatoclostridium*; however, there were no significant differences with the CD+CR and duCR groups (Figure 6C). Then we correlated the above-mentioned genera with indexes related to colitis in all groups. Spearman’s correlation analysis showed that *Citrobacter* was significantly and positively correlated with mucosal thickness, pathological histological score, DAI index, spleen weight, pro-inflammatory cytokine IL1β, and quantification of Ki67 positive cells, while *Lactobacillus* was substantially and negatively correlated with these indices. In addition, *unclassified_f_Lachnospiraceae*, *norank_f_ Lachnospiraceaeand*, and *Enterorhabdus* were also negatively correlated with these colitis-related indicators; in contrast, *Erysipelatoclostridium* was in positive correlation with these indicators, although there was no significant relationship between these bacteria and these indicators (Figure 6F).

### 3.7. Effect of beCR and duCR on Key Microbial Phylotypes

To further identify microbial taxa that supplement the BL community biomarkers before and during CR infestation, we performed liner discriminate analysis (LDA) in conjunction with effect size measurements (LEfSe) with an LDA score > 2. Cladogram analysis revealed that the levels of specific bacterial taxa varied from phylum to genus within each category (Figure 7A). The LefSe analysis results demonstrated that the CD+CR group mice displayed a higher abundance of *Citrobacter*, and *Lactobacillus* was more abundant in the duCR group. *Enterorhabdus* and *Mucispirillum* were found in greater abundance in the beCR group (Figure 7B). 

## 4. Discussion

The implications of rising global IBD prevalence and incidence for human public health are fundamental and challenging [2,3]. Thus, it is desperate for a concerted effort in disease prevention and health-service innovation to reduce the global burden of IBD. Accumulating evidence has demonstrated that dietary intervention, a primary element in regulating gut microbiota, is effective in the clinical remission and therapy of IBD [6]. However, the mechanism and role of dietary intervention remain elusive. Li et al. found that BL as a dietary supplement significantly improved chemical-injury-induced colitis and colorectal cancer [22,24]; and Tian et al. found that barley leaf insoluble dietary fiber improved dextran sulfate sodium (DSS)-induced colitis by modulating intestinal flora [23]. Our previous study suggested that BL could effectively ameliorate CR-induced colitis (unpublished), and to further elaborate on how BL exerts its ameliorative effect, in this experiment, we supplemented BL before and during CR infection in mice, respectively. Our experimental results indicated that the effect of BL supplementation before infection on the CR-induced colitis was superior to that of BL supplementation upon infection, and this improvement was comparable to that of BL supplementation throughout, suggesting that BL mitigates CR-induced colitis to a large extent in a preventive manner. The dosage of BL utilized in this study is a human equivalent dose that has been reported without observed adverse effects [26]. Our study highlights the potential use of BL as a nutritional supplement that is suitable for the clinical management of IBD in humans.

CR is a mouse natural A/E bacterium that causes severe colitis when infected with C3H/HeN mice [13]. Consistent with previously reported studies [14], CR resulted in reduced body weight, increased DAI index, diarrhea, and a shortened colon in mice. Our results indicated that BL supplementation during infection was not effective in improving CR-induced colitis symptoms, whereas BL supplementation before infection could significantly ameliorate CR-induced colitis and produced similar ameliorative effects to those of BL supplementation throughout (Figure 1B–E). CR infestation achieves extraintestinal transmission through the lymphatic circulation, resulting in splenomegaly. Our results suggested that supplementation with BL before CR infestation significantly inhibited the extraintestinal spread of CR, which might be related to its ability to reduce the CR burden in mice (Figure 4A–C). Therefore, dietary supplementation with BL prior to CR infestation significantly alleviated the CR-caused splenomegaly (Figure 1F). Pathogenic bacteria or exogenous agents would induce a host immune response, so we detected the content of inflammatory cytokines in mouse colonic tissue. The results showed that pre-infestation dietary supplementation with BL significantly suppressed the elevation of pro-inflammatory cytokines TNF-α, IL-1β, and IFN-γ caused by CR, suggesting that BL supplementation before CR infestation improved splenomegaly associated with a decrease in circulating pro-inflammatory cytokines (Figure 3). In contrast, supplementation with BL during CR infestation was less effective in improving colitis caused by CR than supplementation with BL before infection.

CR colonization of the intestine could trigger a series of intestinal injuries. TMCH, the hallmark feature of CR-infected mice, is a consequence in which the CR utilizes its type III secretion system to inject virulence factors into enterocytes, resulting in an overproliferation of immaturely differentiated cells, and is characterized by crypt hyperplasia, apical enterocyte surface heterogeneity, and mucosal thickening [13]. In addition, CR can destroy host mitochondria, thus increasing the oxygen content in the intestinal lumen and accelerating its proliferation to compete with the commensal intestinal flora [27]. CR initially colonizes the cecum patch to adapt to the in vivo environment and then migrates to the distal colon. Our results indicated that pretreatment with BL significantly ameliorated CR-induced colonic crypt hyperplasia, which was associated with a reduction in the excessive proliferation of Ki67 cells (Figure 2A,E). In addition, dietary supplementation of BL prior to infestation also ameliorated CR-induced colonic and cecal inflammatory cell infiltration, ulceration, and loss of epithelial integrity (Figure 2A–C and Appendix A). The goblet cell, a mucus-producing cell, is an important component of the intestinal mucosal barrier. Previous studies have found that Muc2 synthesis is essential for host protection during A/E bacterial infection because it reduces the overall number of pathogens and symbionts associated with the colonic mucosal surface [28]. Our data revealed that dietary supplementation of BL prior to infestation was effective in ameliorating the reduction of goblet cell numbers caused by CR (Figure 2A,D), and this partially explained why it suppressed CR load in feces and other tissues of mice (Figure 3). However, the improvement of intestinal histopathology by BL supplementation at the time of CR infestation was far inferior to that of BL supplementation before infestation (Figure 2).

It has been observed that CR causes a profound gut microbiota dysbiosis which is defined by an overgrowth of CR and a decline in the richness and general diversity of the resident microbiota [14]. An alpha and beta diversity analysis of the bacterial community showed that pre-infection dietary supplementation of BL significantly ameliorated the reduced richness and diversity of the intestinal flora and improved gut microbiota dysbiosis with similar flora composition and structure to that produced by feeding BL throughout the experiment (Figure 5). Multiple studies have demonstrated an elevated proportion of Proteobacteria in the gut microbiota of IBD patients [29]. Proteobacteria exploits the host’s immune system to facilitate the recruitment of pro-inflammatory cells, leads to dysbiosis, and ultimately promotes the progression of IBD [29]. Our data suggested that the dietary supplementation of BL prior to infestation could significantly suppress the relative abundance of Proteobacteria, while increasing the content of Firmicutes (Figure 6A,B). At the genus level, pretreatment with BL supplementation significantly suppressed CR levels, and this result is consistent with the results of reducing the CR burden in feces (Figure 4A and Figure 6C,E). Interestingly, dietary supplementation with BL, either before or during infestation, could enhance the relative abundance of *Lactobacillus* (Figure 6C–E), indicating that BL might act as a prebiotic to enrich *Lactobacillus*. Numerous studies have demonstrated that *Lactobacillus* could improve CR-induced colitis through competitive inhibition, replacement rejection, and moderate modulation of the Wnt/β-catenin signaling pathway to avoid over-activation [30]. Using in vivo fluorescence imaging experiments, Waki et al. found that single feeding of *Lactobacillus brevis KB290* could reduce the susceptibility of mice to CR infection [31]. In another study, pretreatment of C57BL/6 mice with *Lactobacillus rhamnosus GG* (LGG) improved CR-induced colitis in a TLR2-dependent manner [32]. *Lactobacillus acidophilus* could reduce the diarrheal phenotype of CR infection in mice [33], and early pre-inoculation with *L. acidophilus* could enhance host defense against CR infection and attenuate colitis [34]. Apart from *Lactobacillus*, dietary supplementation with BL prior to infestation also elevated the relative abundance of *Lachnospiraceae* and *Enterorhabdus* (Figure 6C)*. Lachnospiraceae* is known to perform an essential function in maintaining intestinal health and protecting its hosts. Desen et al. found that angiogenin could mitigate the severity of colitis by modulating *Lachnospiraceae* and α-Proteobacteria [35]. A recent clinical trial has shown that Crohn’s disease patients exhibit a strong adaptive immune response to human-derived *Lachnospiraceae* flagellin, suggesting that *Lachnospiraceae* might be a target for prognosis and future personalized therapy [36]. Another study showed that mice with resected cecum tissue, resulting in a reduced level of *Lachnospiraceae*, exacerbated CR-induced colitis [37]. Noticeably, dietary supplementation with BL prior to infestation suppressed the relative abundance of *Erysipelatoclostridium* besides CR content (Figure 6C). *Erysipelatoclostridium* is a Gram-positive bacterium that causes invasive infections in various tissues, especially in immunocompromised individuals [38]. Manabu et al. found that *Erysipelatoclostridium* enhanced Th1 response after colonization of germ-free mice, leading to colonic mucosal damage in mice [39]. However, the effect of BL supplementation during infestation on the regulation of CR-induced intestinal flora dysbiosis was not as good as that of BL supplementation before infestation. BL powder, rich in dietary fiber, could increase the content of short-chain fatty acids and aromatic metabolites ferulic acid (FA) in the intestinal tract [40]. Hwang et al. found that FA could be used as an intestinal protective agent to improve the intestinal epithelial barrier environment [41]. Moreover, it was shown that the intraperitoneal administration of butyric acid significantly improved CR-caused colitis [42], and Wang et al. found that butyric acid improved intestinal immune homeostasis by enhancing IL-22 production [43]. Niacin, a fermentation product of intestinal flora, ameliorates colitis and colorectal cancer in a Gpr109a-dependent manner [44]. Moreover, our previous studies have demonstrated that the intestinal flora metabolites of BL, namely inosine and secondary bile acids, also could ameliorate DSS-induced colitis [23,24]. Thus, more research is needed to identify the individual *Lactobacillus* species and metabolite that contribute to the protective benefits of BL against CR-induced enteric infection and to examine the underlying mechanisms.

## 5. Conclusions

In this experiment, our results revealed that pretreatment with BL could dramatically improve CR-induced body weight loss and attenuate CR-induced intestinal damage; it also significantly reduced host CR burden and improved CR-induced gut microbiota disorders (Figure 8). However, the effect of BL supplementation during infestation on the improvement of CR-induced colitis was not as good as that of pretreatment with BL. In conclusion, BL improvement of CR-caused colitis might be largely in a preventive manner. Our works provide a reference for the therapy and prevention of IBD.

## Figures and Tables

**Figure 1 nutrients-14-03833-f001:**
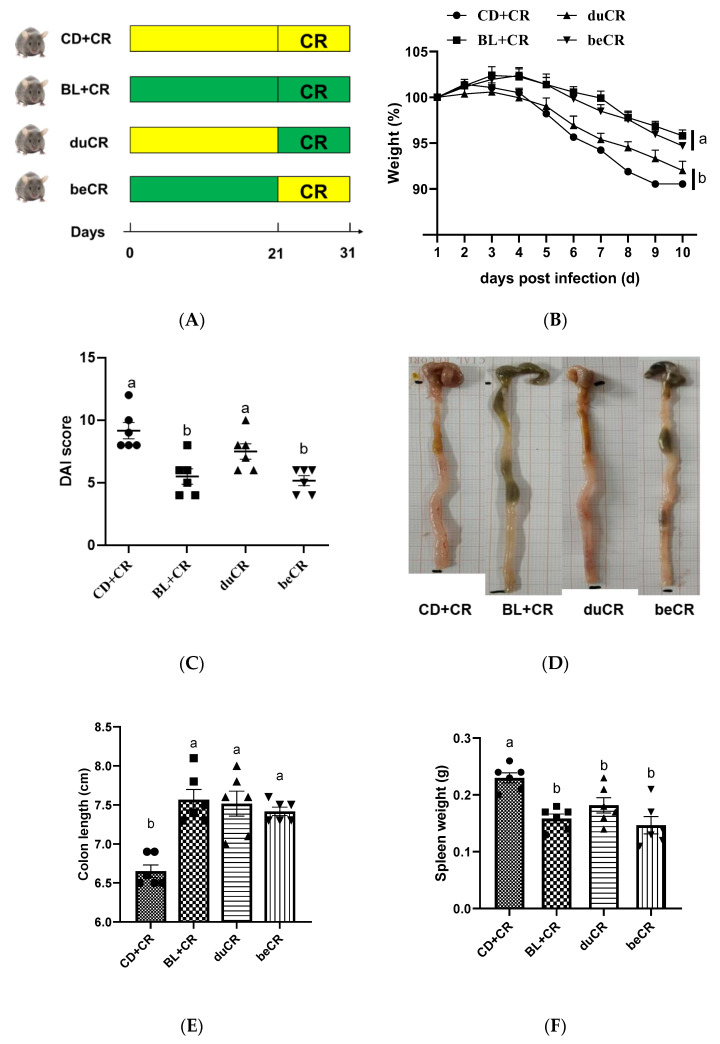
Effect of BL supplementation on mice before and during CR infestation. (**A**) Study design of mice grouping and treatment (*n* = 6 per group). (**B**) Weight loss. (**C**) Disease activity index (DAI) scores. (**D**) Representative images showing the gross appearance of the intestinal tissues; (**E**) Colon length. (**F**) Spleen weight. The a, b means in the same bar without a common letter differ at *p* < 0.05.

**Figure 2 nutrients-14-03833-f002:**
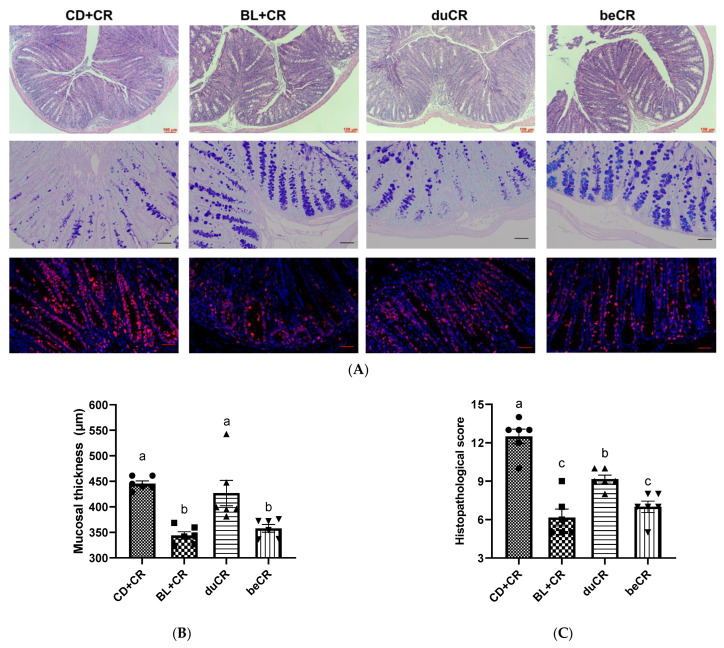
Effect of the beCR and duCR intervention on improving colonic histopathology. (**A**) Representative H&E staining images of colon tissues. Scale bar, 100 μm. Representative Alcian blue staining images of colonic sections. Scale bar, 100 μm. Representative Ki67-stained immunofluorescence images of colon tissues. Ki67 immunofluorescence is indicated in red, and DAPI nuclear staining is blue. Scale bar, 50 μm. (**B**) Mucosal thickness. (**C**) Histopathology score of colonic section. (**D**) Goblet cell positive density. (**E**) Quantification of Ki67 positive cells. The a, b, c means in the same bar without a common letter differ at *p* < 0.05.

**Figure 3 nutrients-14-03833-f003:**
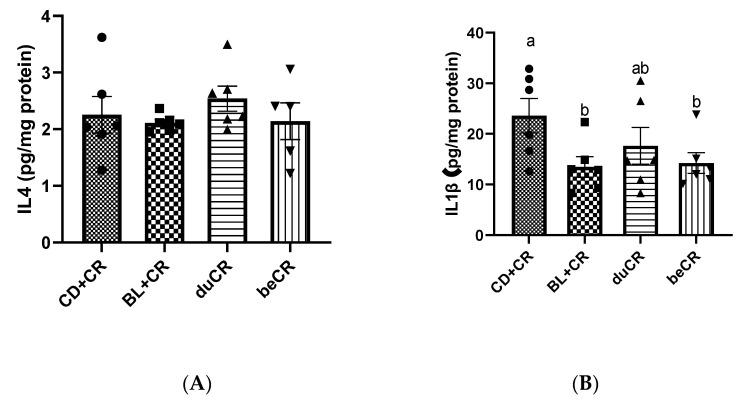
Effect of the beCR and duCR intervention on inflammatory cytokines in colon. The concentrations of anti-inflammatory cytokines (IL-4) (**A**) and pro-inflammatory cytokines IL-1β (**B**), TNF-α (**C**), and IFNγ (**D**) in colonic tissue of mice. The a, b means in the same bar without a common letter differ at *p* < 0.05.

**Figure 4 nutrients-14-03833-f004:**
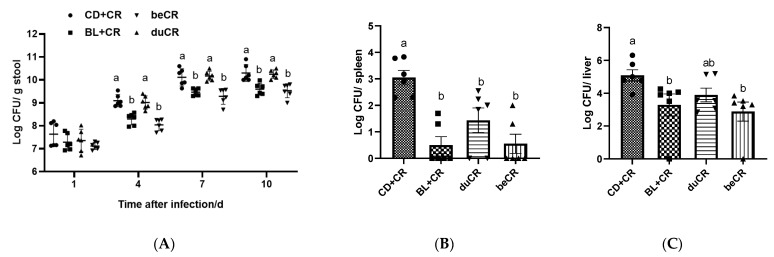
Effect of the beCR and duCR intervention on CR colonization. (**A**) Number of CR in feces on day 1, 4, 7, and 10 p.i. (**B**) Number of CR in spleen on day 10 p.i. (**C**) Quantification of CR in liver on day 10 p.i. The a, b means in the same bar without a common letter differ at *p* < 0.05.

**Figure 5 nutrients-14-03833-f005:**
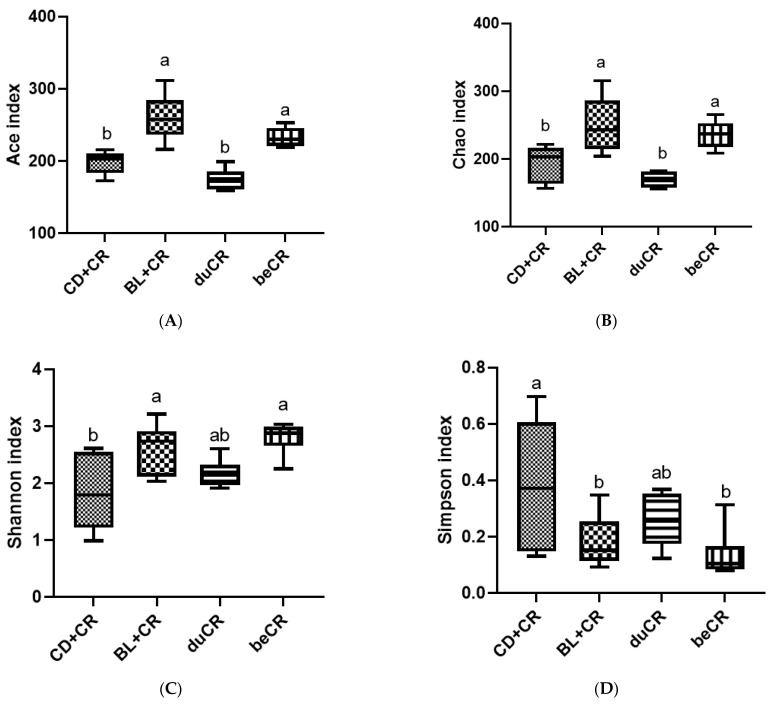
Effect of the beCR and duCR intervention on modulating alpha and beta diversity of intestinal flora. The richness of gut microbiota evaluated by (**A**) ACE index and (**B**) Chao index. The diversity of gut microbiota evaluated by (**C**) Shannon index and (**D**) Simpson index. (**E**) NMDS analysis on OTU level. (**F**) PLS-DA analysis on OTU level. The a, b means in the same bar without a common letter differ at *p* < 0.05.

**Figure 6 nutrients-14-03833-f006:**
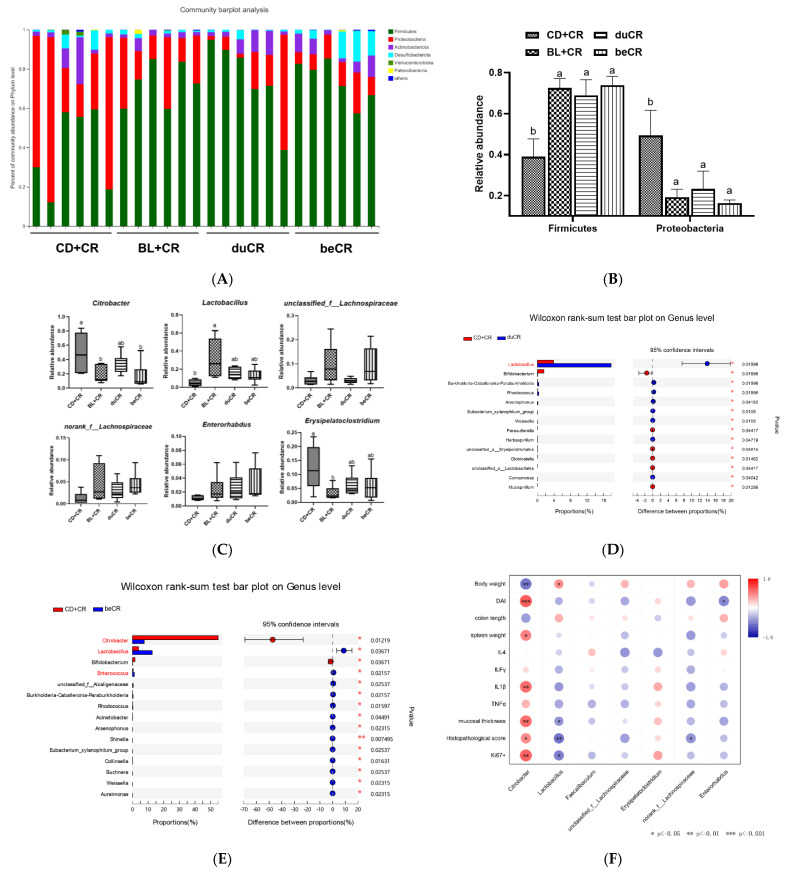
Effect of the beCR and duCR intervention on regulating taxonomic microbial community profiles. (**A**) Taxonomic distributions of gut bacterial composition at the phylum level. (**B**) Relative abundance of Firmicutes and Proteobacteria. (**C**) Relative abundance of *Citrobacter, Lactobacillus, unclassified_f_Lachnospiraceae, norank_f_Lachnospiraceae, Enterorhabdus,* and *Erysipelatoclostridium*. (**D**) Wilcoxon rank-sum test of group CD+CR and duCR at the genus level. (**E**) Wilcoxon rank-sum test of group CD+CR and beCR at the genus level. (**F**) Spearman correlations analysis between the microbiota and colitis-related index. The a, b means in the same bar without a common letter differ at *p* < 0.05.

**Figure 7 nutrients-14-03833-f007:**
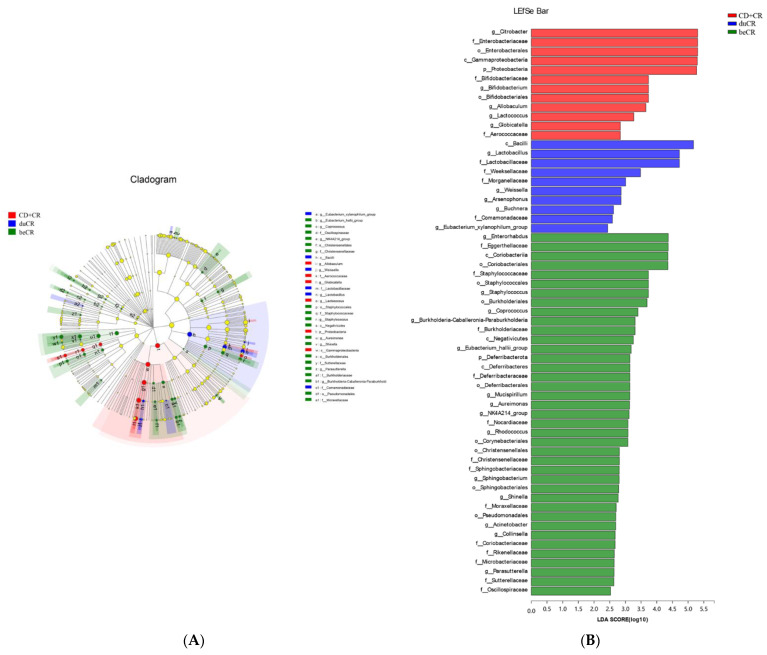
Effect of the beCR and duCR intervention on key microbial phylotypes. (**A**) Cladogram and (**B**) LDA scores derived from LEfSe analysis.

**Figure 8 nutrients-14-03833-f008:**
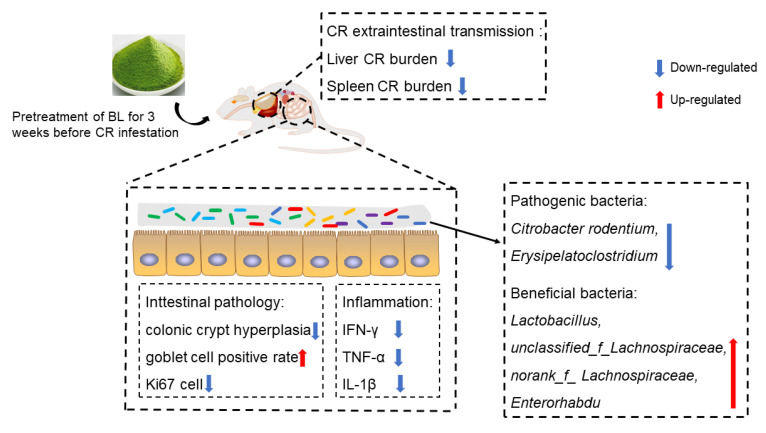
Schematic diagram of BLs’ role in improving CR-causing colitis through a preventive approach.

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
