# Peer review of "Barley Leaf Ameliorates Citrobacter rodentium-Induced Colitis through Preventive Effects"

_nutrients, 2022, doi:10.3390/nu14183833_

Round 1

Reviewer 1 Report

This is an interesting paper about the effect of supplementing the diet with dry powder of barley leaves on Citrobacter rodentium-induced colitis. The work is clearly presented, and the conclusions are consistent with the results.

I am not an expert in the study of the diversity of the intestinal microbiota and I cannot add much on this point, apart from saying that what is explained to me is completely consistent.

On the other hand, it is known that this supplement is very rich in insoluble fiber, as indicated in table S1. Also, that the insoluble fiber of dry barley leaf powder is degraded by the intestinal microbiota to produce a set of short-chain fatty acids, and, also, other aromatic acids, among which ferulic acid stands out. Also, that ferulic acid has positive effects on ulcerative colitis. There are several works that in one way or another deal with the benefits of the insoluble dietary fiber of barley leaves.

In this paper, I miss the correlation between the generation of these metabolites and the changes in the production of pro-inflammatory cytokines and in the microbiota. Studying this experimentally would surely involve a relatively long new work. It is not my goal to ask for it now, but I do consider it essential to thoroughly discuss what has been published about changes in the microbiota and colitis correlated with barley leaf fiber metabolites.

Reviewer 2 Report

Barley leaf ameliorates Citrobacter rodentium-induced colitis through preventive effects

Yu Feng, Daotong Li, Chen Ma, Meiling Tian, Xiaosong Hu and Fang Chen

The inflammatory bowel disease is increasing globally. Despite an abundant literature discussing the pathophysiology and treatment of this disease, there are no current efficacy treatment.

In this works, authors aimed to understand if Barley leaf exerts a prophylactic or therapeutic effect in inflammatory bowel disease. They observed that Barley leaf supplementation prior to infection significantly reduced DAI index, weight loss, colon shortening, colonic wall swelling, and transmissible murine colonic hyperplasia. Barley leaf supplementation also reduced the levels and expression of pro-inflammatory cytokines. The authors conclude that barley leaf can act as a preventive agent against CR-caused colitis.

Although the authors' consent and observations are interesting, there are some points that require further justification.

Major comments:

- A more detailed explanation of the beneficial effects of Barley leaf should be informed, with references. In which diseases were studied?

- A figure explained the experimental protocol should be included. It helps a lot to understand your experiments.

- Figure 1 B, the x axis is days?

- BL reduces colitis mainly by reducing inflammation. What is the effect in anti-inflammatory cytokines?

- Does the Barley leaf ameliorates colitis in other models? A brief discussion should be added.

Minor comments:

- You should explain what is DAI index in the abstract

- In the abstract, please change “Dramatically reduced” for significantly reduced.

- In the methods you should have a more detailed explanation of Disease Activity Index.

- Figure 2A – Please provide the legend for the immunofluorescence (red and blue).

- I cannot find Supplementary figures.

Reviewer 3 Report

This is a clearly written manuscript that can be relevant for the audience to read. I consider that only minor points need to be addressed:

1.     Please add “Disease activity index” to the abbreviation “DAI” in the abstract.

2.     Please add a brief description on how to calculate DAI to methods (section 2.5).

3.     Please describe the sample collection and preparation process in methods section 2.8.

4.     Please include dot plots in all figures as the authors did in Figure 1C or Figure 4A

5.     Please explain in methods or results what does Ki67+ staining mean.

6.     Figure legend 2: the authors use “c” for statistics on the graphs but it is not added to the figure legend. Please add
